# An Assessment of Socio-Economic Systems' Resilience to Economic Shocks: The Case of Lithuanian Regions

**Jurgita Bruneckiene [1,*], Irena Pekarskiene [1], Oksana Palekiene [1] and Zaneta Simanaviciene [2]**

[1]  School of Economics and Business, Kaunas University of Technology, 44239 Kaunas, Lithuania;
     irena.pekarskiene@ktu.lt (I.P.); oksana.palekiene@ktu.lt (O.P.)
[2]  Institute of Economics, Mykolas Romeris University, 08303 Vilnius, Lithuania; zasiman@mruni.eu
[*]  Correspondence: jurgita.bruneckiene@ktu.lt; Tel.: +370-37-300-550

**Abstract:** Various socio-economic systems (countries, regions, or cities) and their economies suffer different kinds of economic shocks. If the system is not resilient, its economy can incur losses. Only the systems whose economies are less vulnerable and/or are able to recover from the economic shock quite quickly are able to ensure economic sustainability, competitiveness, and welfare both now and in the future. The concepts of socio-economic systems' resilience to economic shocks, vulnerability, and recovery, as well as the resilience assessment peculiarities, are all analyzed in this article. The methodology introduced for the assessment of a socio-economic system's resilience to economic shocks consists of two parts: a model of a system's resilience to the economic shock's capacity-related factors (Resilio) and an index of a socio-economic system's resilience to the economic shocks (Resindicis). The Resilio model could be used as a universal methodological framework for analyzing the resilience of socio-economic systems of different levels (countries, regions, or cities). The set of quantitative indicators compiling Resindicis should be adjusted to the specifics of each socio-economic system and the availability of statistical data. Empirically, the methodology was validated on the example of 10 Lithuanian regions (counties). The methodological principles for the assessment of a socio-economic system's resilience are also provided. The main advantages and drawbacks of the methodology are discussed in order for further development and an increase in its practical application.

**Keywords:** resilience; vulnerability; recovery; socio-economic system; economic shocks; region

## 1. Introduction

The upheaval of any socio-economic system causes a small or large effect (commonly losses) and brings about different consequences that negatively affect social and economic welfare, in addition to raising additional barriers for sustainable socio-economic development. For instance, the global economic crisis of 2008–2009 hit all socio-economic systems. Adelson [1] estimated that the losses of this global crisis amounted to nearly 15 billion United States (US) dollars. It should be noted that not all socio-economic systems went through the same economic difficulties. In fact, the economies differed by their vulnerability and recovery rate. Some economies (for example, the United Kingdom, Canada, Sweden, and Australia) were more resilient in comparison with others. The constant change in economic conditions, new challenges, striving for a higher quality of life, and the intense competition among the countries, regions, and cities (for human capital, investment, technologies, and other determinants of economic development) lead to the need for socio-economic systems to recover as soon as possible from economic shock or other challenges. In other words, socio-economic systems

have to be resilient to economic shocks, as only a resilient system can ensure, in both the future and the present, economic stability, competitiveness, and high quality of life. Investment targeted at the promotion of resilience is economically efficient; prevention, quick recovery, and lower losses are more beneficial than a prolonged economic recession and its negative consequences.

In this article, a socio-economic system is understood as the economy of a country, region, or city. The use of this concept enables resilience to be treated as a process, because these systems are characterized by intricacy and complexity, as well as by dynamics. Based on Ginevicius et al. [2], these systems are in constant motion and continual development. In this article, the socio-economic system is understood as a structured set (whole) of entities (enterprises, organizations, communities, or societies) and objects (natural geographic environment, technologies, or infrastructure) interacting in a specific geographic area (country, region, or city). The term "set" excludes many elements from the environment or higher hierarchical system and, at the same time, outlines the limit of the analyzed system. The term "interacting" means that only interconnected elements compose the "set". The term "geographic area" defines the area in which these elements interact. The term "structured" demonstrates both the hierarchical arrangement of the system's elements, as well as their derivatives, and the fact that their interaction contributes to the overall objective of the system.

The problems of the vulnerability of socio-economic systems and their ability to recover after economic shocks currently receives considerable attention from scientists, politicians, and strategists. The necessity for efficient strategies and activity plans to allow the systems to increase their resilience to economic shocks results in much discussion at the strategic and political levels. Although strategists are inclined to focus on resilience to global phenomena such as growing food prices, demographic pressure, climatic changes, natural disasters, environmental degradation, catastrophes, and other challenges, socio-economic systems' ability to withstand economic shocks remains an extremely topical issue.

As an example, in 2014, at the request of the European Parliament, a study entitled "Impact of the Economic Crisis on Social, Economic, and Territorial Cohesion of the European Union" was conducted [3]. The study covered the problems of socio-economic systems' resilience to economic shocks and the issue of their development. In addition, the problems of resilience were indirectly covered in the smart, sustainable, and integral growth strategy "Europe 2020" (priority of sustainable growth) announced in March 2010. It should be emphasized that increasing resilience to economic shocks is a long-term process and a final goal; therefore, the analysis of this economic concept requires extensive, long-term, and in-depth research. Thus, in order to increase the resilience of the socio-economic system to economic shocks and to develop efficient resilience promotion strategies and measures, a comprehensive analysis of the current situation should first be performed (including an evaluation of levels of vulnerability and recovery), and the factors determining the system's resilience and those forming its ability to be resilient should be identified, along with the detection of strengths and weaknesses. The methodological methods that enable the accumulation of accurate and timely information about the socio-economic systems' resilience, and that enable an assessment of the dynamics of resilience and the efficiency of strategies are the main tools of strategic planning and the key determinants of resilience promotion.

Scientists assess a socio-economic system's resilience by employing either a static or dynamic method. The static method allows for comparisons of different systems and ranks them as more or less resilient in a particular period of time [4–8]. Nevertheless, this method does not reflect the dynamics of comparing systems' resilience. The dynamic method includes the time factor and characterizes a socio-economic system's resilience and its ability to recover after an economic shock through managerial, but not economic, factors [9–12]. Nevertheless, the latter method does not provide an opportunity to compare systems' resilience.

Therefore, an analysis of the literature disclosed that, thus far, there is a lack of assessment methodology for socio-economic systems' resilience that would, employing either the static or dynamic approach, allow for a comprehensive assessment of the problems of resilience to economic shocks. The combination of both methods would provide a more comprehensive assessment of and deeper

insight into a socio-economic system's resilience to economic shocks in a time perspective, as well as evaluate the level of vulnerability and recovery.

The lack of a complex methodology for the assessment of a socio-economic system's resilience to economic shocks is one of the main obstacles to assessing the ability of the system to resist economic upheavals; it also impedes the development of efficient resilience promotion strategies.

*The purpose of this article* was to identify the main factors of socio-economic systems' resilience to economic shocks and to use the index evaluation method for the assessment of socio-economic systems' vulnerability and recovery from the economic shock.

*The research methods* involved a systematic, comparative, and logical analysis of the scientific literature based on the methods of comparison, classification, systematization, and generalization; mathematical and statistical data processing; and multifaceted evaluation methods.

## 2. Literature Review

### 2.1. The Concept of a Socio-Economic System's Resilience to Economic Shocks

The concept of socio-economic systems' resilience to economic shocks is one of the most complicated and complex research areas. This concept is treated as a multifaceted notion [13,14]. It is related not only to the variety of economic shocks and their effects, the broadness and versatility of the concept itself, and the abundance of the factors of resilience, but also to different scientific attitudes toward resilience: as a weakness [15–17] or an advantage [18,19], static and dynamic views to this concept. The different systems' specificities [20] also matters. In the research on infrastructure resilience, the concept is usually analyzed by equilibrium and in a static view. In research related to ecological and economic resilience, the socio-economic system's reaction to shock is analyzed in both static and dynamic views. Martin (2012) [21] used a complex adaptive systems theory for analysis of regional economic resilience. Terms such as adaptability and transformability [22], flexibility [23,24], reaction [20,25], response [26], and ability [23], which characterize the resilience concept, represent more the dynamic than the static view to the problem.

An extended approach to the concept of resilience—spatial economic resilience—noticeably appeared in the latest research [13]. The novel concept combines resilience of the economy and its actors, ecology, and infrastructure into one general system. The uniqueness of this approach is that the resilience of the socio-economic system is determined not only by the resilience of the environment and infrastructure, but also by the overall resilience of the socio-economic actors (community, enterprises, organizations, and authorities at different levels), i.e., the ability of these actors to be resilient and to enable the environment and infrastructure for the whole system's resilience. This resilience approach is used to both further analyze the problem and develop a methodology for the resilience assessment.

Researchers [4,15,21,25,27] highlight three main distinctive qualities of a socio-economic system's resilience as follows:

(a) shock—the cause of the changes in the socio-economic system;
(b) vulnerability—the sensitivity of the socio-economic system or the extent to which it is affected by shock;
(c) recovery—the ability of the socio-economic system to react to (i.e., resist, adapt to, absorb, or overcome) an economic shock by following a particular strategy.

Scientists both analyze different types of shocks (e.g., 2008 global economic crisis, Ukrainian–Russian conflict, refugee flows to Europe from Africa and the Middle East, etc.) and try to find the causes of their occurrence (e.g., natural disasters, political unrest, changes in the demographic situation, technological progress, etc.). The causes of each economic shock are different. Hennessey et al. [28] named the impact of the globalization process as one of the main causes of the 2008 global economic crisis. Rakauskienė and Krinickienė [29] identified the growth of the role of capital markets, the growth of the uncertainty of financial markets, the intensity of the globalization processes,

social inequality, the lack of moral and ethical values, the inadequate supervision and irresponsible activity of financial institutions. Martin [21] described the increased inconsistency of production capacity to consumption and purchasing power, excessively weak legal regulation and control of the stock market, and extended distrust to the nearest economic future. Forasmuch, due to the abundance and diversity of the economic shocks, as well as the causes of their occurrences, it can generally be concluded that there is no unambiguous definition of an economic shock, because it affects different objects in different intensities, and differs in its operational conditions, exposure time, specifics, and environmental factors.

In order to assess an socio-economic system's resilience to an economic shock, the authors of the article propose the following definition of an economic shock: *an economic shock is an unplanned change in operational conditions, or economic, politic, social, and/or natural environment; it is a phenomenon or an event in regional, national, and/or international economics which, if disregarded or managed with consideration of the current development strategy, may determine a sudden and significant negative and/or possibly positive impact on a system's development.* It should be noted that, although scientific and strategic documents mainly focus on the negative impact of economic shocks, the attitude that environmental changes provide new opportunities may reveal the positive effects that the economic shocks might have on the development of a socio-economic system's economics.

The term "vulnerability" is closely connected with the resilience concept. Usually, vulnerability illustrates the degree to which a socio-economic system is susceptible to harm or its sensitivity to particular shocks. Scientific literature [13,30,31] emphasizes that there is no clear link between vulnerability and resilience in the context of the spatial economic resilience concept. There are several reasons for this explanation. Firstly, both theoretical and empirical studies show different relationships between these two concepts. As an example, a highly resilient system can have both high and low vulnerabilities. Briguglio [4] stated that even an economically successful country can be extremely vulnerable, and called such an economic situation "the Singapore paradox". Turvey was of the opinion that small economies are more vulnerable in comparison with large economies [17]. Cainelli et al. (2018) [32] found that regions with high levels of technological relatedness show high levels of resilience. Pendall et al. (2012) [33] treated a more resilient system as a system with less vulnerable sub-systems. As noted by Baldacchino and Bertram, the stereotype that small economies are extremely vulnerable already resulted in merciful policies whereby small economies are granted different discounts, privileges, and financial support [19]. On the other hand, Easterly and Kraay argued that small economies experience less destructive economic shocks than large economies [20]. Graziano and Rizzi (2016) [31] found a split between the north and south of Italy in the economic sphere from a map of vulnerability and resilience. Another reason is the lack of research about reducing economic vulnerability factors; however, more attention is paid to the factors determining the overall economic resilience. Modica et al. (2018) [30] noted that only a few papers focus on economic measures of vulnerability, even though the economic and the social environment are fundamental aspects evaluated in all studies concerned.

Despite the fact that scientists [30,31] found some common and different characteristics between vulnerability and resilience, mainly regarding the socio-economic conditions of the objects of the analysis, the analysis of vulnerability vs. resilience is still experiencing a theoretical/methodological gap—there is a different scientific approach to the vulnerability and the specificity of the shock itself. In most cases, scientists [13,14,31] treat vulnerability as an inevitable situation of the socio-economic system caused by exogenous shocks. Such an approach appeared due to the specifics of infrastructural and ecological resilience, whereby shock is an external and uncontrollable factor or condition. Modica et al. (2018) [30] noted that most analyses of vulnerability refer to natural disasters, which can be conditionally or totally controlled. Resilience is usually interpreted as the ability to recover from shock or to reach the pre-shock level. Economic vulnerability was treated by Briguglio et al. (2009) [14] as an inherent condition that affects the exposure of a territory to exogenous shocks. Economic resilience was treated by these authors as the set of actions implemented by socio-economic systems to help the

territory recover from and/or to adapt to a negative shock or to help benefit to the greatest extent from a positive shock. Taking into account the dynamics and vitality of the socio-economic system and the specifics of economic shock itself, the authors of the article expanded the dominant approach of the vulnerability concept in the scientific literature by analyzing spatial economic resilience. Under this approach, the vulnerability could be treated as (a) an inherent condition, or (b) a (partly) managed process where a socio-economic system can be prepared for economic shock, whereby it can be partially reduced and managed. This approach clarifies the concept of spatial economic resilience as the sum of vulnerability and recovery and the ability of the system to manage both vulnerability and recovery.

The term of a socio-economic system's *recovery* after an economic shock is often explained through the concept of an "equilibrium state", whereby a system's resilience to economic shocks is treated as the ability of its economy to return to its pre-shock condition or to reach pre-shock economic rates [34–38]. In this regard, the focus is on the time and speed at which its economy returns to its state of equilibrium. With reference to the concept of "equilibrium state", the assessment of a system's resilience should include the assessment of its ability to either to return to its pre-shock condition or to develop a new equilibrium (the dynamic aspect).

Socio-economic systems' resilience to economic shocks is a process during which an economic shock can lead to the changes in a system's economic structures and functions, which, in turn, may affect the strategies of its development after the shock. The literature [20,25] proposes four interrelated strategies for a system's development (the so-called development trajectories): *resistance; recovery; reorientation; renewal.* Thus, depending on the causes and duration of an economic shock, along with the depth of the damage, the possession of resources, and the ability to exploit these resources, the reaction of a socio-economic system to an economic shock can be twofold. In one case, when economies are able to resist an economic shock and adapt to a new situation, the damage is not so serious and does not have any significant impact on the development of the economies (an economically resilient system). In the other case, when economies face a larger shock, two scenarios are possible. (1) The socio-economic system, which is able to exploit its resources and opportunities purposefully and efficiently, recovers after an economic shock or selects a new way of economic development—renewal or reorientation (a resilient system). (2) A socio-economic system, which is not able to resist an economic shock, suffers heavy losses and the development of its economy is disrupted (a non-resilient system).

With the aim of specifying the term "recovery after an economic shock" and incorporating Martin's development strategy of "recovery", this article defines "recovery after an economic shock" as an "operation without changing the substance", having in mind an economic shock that does not result in any significant changes to the socio-economic system's economic structures and functions [20]. It is argued in this article that a system can reach its pre-shock condition by resisting, reorienting, renewing, or operating further without any substantial changes, i.e., socio-economic systems' economies can recover without any significant changes in their economic structures. Previous studies revealed that different recovery strategies are not universal, as systems differ by their specifics, and economic shocks differ by their nature. Thus, every socio-economic system has to develop a separate target strategy.

In summary, it can be stated that the response of a socio-economic system to economic shock, i.e., the ability of the system to resist in the time perspective, manifests through its capacity to withstand external pressures, keep its economic development uninterrupted, exploit the capacity for a positive reaction to external changes, adapt to the changes in the long run, and learn from past experience. This attitude supports the statement that a socio-economic system's resilience to economic shock is a strategy of its economic development, while the evaluation of changes in its economic development is a reliable measure of its resilience. A review of the literature enabled a formulation of a new definition of a socio-economic system's resilience to economic shock. Further analysis and evaluation of this concept is based on this. *A socio-economic system's resilience to economic shocks includes both the interrelated abilities and possibilities of its economic entities to use its dynamic capacities and infrastructure, maintain the expected development of its economy now and in the future, be left intact or else affected in the least possible*

*manner by an economic shock, and subsequently reach the previous state of the economy before the economic shock in the shortest time by implementing a recovery, renewal, or reorientation strategy.* This definition enables the analysis of resilience as a dual concept—the sum of vulnerability and recovery concepts—i.e., resilience can be increased through a reduction in vulnerability and a strengthening of recovery.

*2.2. The Main Factors of a Socio-Economic System's Resilience to Economic Shocks*

The essence of the concept of a socio-economic system's resilience to economic shocks is characterized not only by the definition, but also by the factors determining it. It should be noted that one factor does not fully reflect the difficulties inherent in the system's resilience to internal and external shocks; therefore, an increasing number of different factors were identified in the scientific literature. Most often, scientists distinguished the following factors of the socio-economic system's resilience to internal and external shocks: diversity of resources, labor force surplus, stability, ingenuity, adaptability, flexibility, cooperation, interdependence and support, autonomy, networking, and innovation [23,30,31,37,39–42]. Based on Chuvarajan et al. [43], diversity of resources is one of the main features of all complex adaptive systems, which helps ensure the stability of the economy if disturbance occurs in any branch of industry. Van der Veen and Logtmeijer [26] highlighted surplus as the main factor for increasing resilience and identified with it the ability of the system to respond to shock, overcome dependencies, and use substitutes, or even move production to a new location. According to Walker et al. [22], adaptability is understood as the ability of the entities in the system to positively influence its resilience. Godshalk [23] equated adaptability with the ability to learn from experience and flexibility. These authors agreed that innovation is essential for a system's resilience. When assessing the experience component, being able to learn from the past shocks is emphasized in order to anticipate and deal with possible future shocks. Walker et al. [22] stated that the ability to change or create a completely new system is one of the main factors of a system's resilience. Creativity allows for the achievement of a higher level of performance by adapting to new circumstances and learning from the experience gained from past shocks [40]. According to researchers, creativity is a way to cope with dangerous events that cannot be foreseen, or that are of a higher magnitude than expected. Flexibility allows the system to recover from the negative effects of the shock [24]. For Godshalk [23], the flexibility of change determines the ability to adapt. The speed or time of returning to the equilibrium state and the speed of accessing and using resources were indicated as the core factors of a system's resilience. Fiksel [44] and Chuvarajan et al. [43] emphasized the importance of networking in order to ensure a system's resilience. Researchers argued that only a system's relations and unifying forces determine the functioning of the system.

It should be emphasized that a socio-economic system's resilience to economic shocks is not sufficient to identify its determining factors. Based on Hudson [45], the importance and the change of each factor over time in each system are different. Appropriate processes, structures, and conditions, which would contribute to the timely implementation of policies and strategies, must also be applied [46]. Researchers emphasized that the success of an economy's development in the changing environment is increasingly determined not by static factors and resources, but by dynamic capacities, which involve the ability to cope with a rapidly changing environment [9,10]. Therefore, it is appropriate to analyze the factors through the capacity for resilience of the socio-economic system. Summarizing the different opinions and studies, this article suggests that the determining factors of a socio-economic system's resilience can be divided into six groups: (1) insight capacity, (2) the socio-economic system's governance capacity, (3) knowledge and innovation capacity, (4) learning capacity, (5) networking and cooperation capacity, and (6) infrastructure capacity.

Insight capacity allows for proactive foresight of opportunities for development, economic shocks, or other threats and problems, and to flexibly react and properly prepare and, if necessary, to form a recovery, renewal, or reorientation strategy. Insight capacity includes strategic insight (perception of development tendencies, and the identification of environmental factors and their changes, and the

possible impact on the development of the socio-economic system, i.e., qualitative dimensions) and economic viability, which can be expressed using quantitative dimensions.

A socio-economic system's governance capacity enables the proper organization and timely implementation of strategic and/or structural changes. This capacity creates the preconditions for overcoming, enabling, and coordinating the whole socio-economic system, together with enabling all economic entities to be resistant to economic shocks. In order to assess the expression and specificity of governance capacity, two main criteria of this dimension are distinguished: government efficiency (including the competence, timeliness, and transparency of economic entities' cooperation), and financial possibilities (reflecting the financial power of the socio-economic system's government and its ability to handle financial resources).

Knowledge and innovative capacity enables the use of knowledge and innovation for economic value creation, as well as assisting in the preparation for a new economic shock or its avoidance and recovery from it. Research and innovation-based solutions allow the achievement of a more efficient (more productive, faster, safer, less expensive, etc.) and more attractive solution to the problem, which is especially relevant during shocks or in order to avoid economic shocks. In order to describe this capability, two dimensions are distinguished: research and innovation (business and government sector investment into research and innovation, an active cooperation between science and business) and an innovation-stimulating environment (a functioning innovation system and positive attitude toward research and innovation).

Learning capacity, in the context of a socio-economic system's resilience to economic shock, is understood as the enabled ability of the system's entities to be interconnected and make opportunities to continuously learn and develop personal and collective competence. Already obtained and newly gained knowledge, competences, and experience during the learning process will be used to create economic value and ensure economic development now and in the future in order to avoid economic shocks and, if necessary, to form a new recovery, renewal, or reorientation strategy based on gained knowledge, competence, and experience. When analyzing learning capacity, two groups of factors are proposed: the learning system (developed systems of science and education and lifelong learning and continuous improvement, and the identity of a knowledge of the socio-economic system), and labor market flexibility and competence (a qualified innovative and entrepreneurial workforce and inquisitive employees in the socio-economic system with a high willingness to learn).

Networking capacity enables the combination of different knowledge, competencies, resources, and opportunities, as well as cooperation ability, to implement, coordinate, lead, and manage collective activities and perform solutions in order to prevent an economic shock or eliminate its consequences. Research showed that evaluating networking and collaboration capacity using statistical indicators in the context of a socio-economic system is a particularly difficult task. In order to more precisely assess the resilience to economic shock, networking and cooperation capacity is assessed in this article using quantitative factors. When analyzing networking and cooperation capacity, two groups of factors are proposed: an established cooperation and feedback mechanism between government and business, and an integration into international and national value, generating chains and networks.

Infrastructure capacity enables the efficient, timely, and flexible use of the dynamic capacities of the socio-economic system. In the analysis of infrastructure capacity, two groups of factors are proposed: a modern and efficient infrastructure system (development of the information and communication technology network, access to the socio-economic system, energetic independence) and sustainability (implementation of the sustainable development principles, the system's tourist attraction, the system's pollution).

Research showed that the analysis of one factor does not reflect the problem of resilience to economic shock. In order to perform a detailed analysis of this problem, it is necessary to systematically analyze the interrelated determining factors of all socio-economic systems entities' resilience to economic shocks. It is important to emphasize that, sometimes, the impact of one or more factors may be negative; however, the socio-economic system may remain resistant to economic shock. In this

case, the negative impact of one or more factors is possibly compensated for by the positive impact of others factors. However, resilience to economic shocks is determined by a variety of interrelated factors which all have different impacts on the overall socio-economic system's resilience; therefore, this problem should be investigated in a complex way.

*2.3. Methods for the Assessment of a Socio-Economic System's Resilience to Economic Shocks*

The assessment of a socio-economic system's resilience is not a novel topic in the literature. Therefore, different resilience assessment methods are defined. Some authors [6,47–49] assessed resilience problems by employing the dynamics of separate indicators, while others [39,50] analyzed best examples and applied econometrics (regression equations) [51] or shift-share and input–output models [52]. Recently, researchers paid increasing attention [53,54] to the problem of structural reforms' impact on a socio-economic system's resilience to economic shocks. The areas of analysis were comprehensive, and included the following: the relationship between structural reforms and economic resilience and vulnerability [55–57]; identification of economic indicators, and forewarning about economic shock [58]; identification of the relationship between economic cycle, gross domestic product (GDP), and resilience [25]; and the comparison of different systems' resilience levels and resilience promotion strategies [59]. The analysis of resilience also covered the relationship between the specifics of some industries and their resilience or vulnerability [30–32]. Meanwhile, other researchers assessed resilience by employing indices [4,6–8,31,37].

Resilience assessment methods presented in the literature often accept the analysis of large economies such as the United States, Germany, Great Britain, Turkey, Greece, Russia, Brazil, China, India, or groups of countries (Organization for Economic Cooperation and Development (OECD), European Union). Despite the fact that hierarchically lower systems are significantly more economically dependent on other socio-economic systems, less attention is paid to their specifics. In addition, assessment methods accepted in the literature do not focus on the assessment of the resilience concept as a component of vulnerability and recovery. Most research methods are used for the determination of the level of a socio-economic system's resilience, the trajectories of resilience movement, or the impact of certain factors on resilience.

Considering the fact that resilience is treated as a multifaceted concept, most researchers proposed assessing resilience by employing indices, and this method is recognized as an appropriate tool for the complex analysis of the problem. Publications focusing on the issues of resilience, as well as the reports announced by the European Commission and the United Nations, propose more than 20 different resilience assessment indices. The most common indices include the following: *resilience capacity index* [60]; *economic resilience index* [36]; *socio-economic resilience index* [47]; *prevalent vulnerability index* [5]; *composite vulnerability index* [7]; *risk reduction index* [6]; *post-2015 indicators for disaster risk reduction* [48], and *resilience cost approach* [49]. It should be noted that the most popular indices are designed for the assessment of the resilience of large economies (I, II level of Nomenclature of Territorial Units for Statistics (NUTS I, II)), which limits their applicability while assessing the lower hierarchical level of socio-economic systems' resilience (NUTS III level).

An analysis of resilience indices showed that they differ through their identification of the factors determining the resilience and their inclusion into the structure of the index itself; however, the stages of their formation remain the same. In general, the indices are structured in the following sequence: (a) methodological justification for identifying the factors of resilience and the selection of indicators for characterizing these factors; (b) rationing of the factors' indicators values; (c) determination of weight coefficients to factors or their groups (researchers classified this stage as one of the most complicated, because the index value and rank directly depend on the presence of weight coefficients, their values, and the method of setting coefficients itself; it is advisable to check how the estimated index and rank vary using different methods for setting weight coefficients); (d) index calculation; (e) index reliability analysis. The most common robustness and sensitivity analysis methods are used.

Despite a wide variety of resilience *assessment models* (which structurally incorporate different factors in one common system) and *methods* (which allow for an assessment of the problems of resilience using *the dynamics of particular indicators*, interconnected relations, impact of separate indicators of resilience, *best examples*, and *resilience indices*), the assessment of a socio-economic system's resilience is a complicated process. Despite resilience concept assessment possibilities and analytical spectrum diversity, researchers came to the unanimous opinion that their applied assessment methods had some limitations because of the specifics of socio-economic systems, the variety of resilience determining factors, economic cycles, the impact and extent of the shock, the diversity of applied political reforms, and others conditions. Therefore, it is recognized that various methods for assessing resilience are possible (if they enable the achievement of the research objective), but their methodological justification is of particular importance.

## 3. Data and Methodology

A socio-economic system develops in a dynamic environment, and is constantly affected by and especially dependent on both changing external and internal factors and the environment, which justifies the necessity for a dynamic approach in order to assess resilience to economic shocks. A complex analysis of resilience to economic shocks and the methods of its assessment enables the identification of the following statements, on the basis of which the methodology for the assessment of a socio-economic system's resilience to economic shocks is formed:

- Resilience can be treated either as a process or a condition. In terms of assessment by an index, resilience is treated as a condition observed over a particular period of time. While analyzing the resilience, along with the ways and measures of its achievement at different periods of time, it is treated as a process;
- Resilience depends not only on the resilience of the system's subjects, but also on the resilience of the system's objects and geographical location;
- Resilience is determined by a set of factors rather than by a single isolated factor;
- Resilience is affected by the factors of internal and external environment;
- The same economic shock affects socio-economic systems in different ways;
- The impact of an economic shock on a socio-economic system's development may manifest at different periods of time: immediately and in the long run;
- Resilience is the sum of vulnerability and recovery;
- Resilience is often characterized by vulnerability depth, recovery time, and duration.

The methodology for assessing a socio-economic system's resilience to economic shocks consists of two parts: the model of capacity-related factors of a socio-economic system's resilience to economic shocks (hereafter called the Resilio model) and the index of a socio-economic system's resilience to economic shocks (hereafter called the Resindicis model). It should be emphasized that the Resilio model is based on the scientific analysis of the concepts of resilience and economic shocks and could be used as a universal methodological framework for analyzing the resilience of socio-economic systems of different levels (countries, regions, or cities). This model is characterized by its universality for both theoretical and empirical research. Meanwhile, the structure of a socio-economic system's Resindicis model can vary (i.e., the set of quantitative indicators) depending on the availability of and possibilities to gather the statistical information, despite the fact that the stages of the Resindicis calculation are universal. In various socio-economic systems, due to the availability of statistical data, different numbers of indicators or different indicators for characterizing the same factor can be used. Therefore, each author of empirical research should develop a new set of indicators characterizing the factors determining the resilience to economic shocks in the Resilio model.

The determination of and beginning of an economic shock plays a key role in the assessment of resilience. Therefore, the identification and justification of an economic shock is identified in Phase 1

of the resilience assessment using an index. Quantitative indicators for selected factors determining the resilience in the Resilio model are identified in Phase 2.

Considering that the possibilities of obtaining statistical data in each socio-economic system are different, and that Lithuanian regions (counties) are analyzed in this empirical research, Table 1 presents a template set of indicators, based on the availability of Lithuanian statistical information at the regional level.

**Table 1.** The factors and quantitative indicators determining a socio-economic system's resilience.

| Resilience Capacity | Quantitative Indicators Defining the Capacity |
| --- | --- |
| 1. INSIGHT CAPACITY<br>*1.1. Strategic insight of the economic subjects*<br>*1.2. Economic viability* | Gross domestic product (GDP) per capita, by purchasing power standards;<br>Share of the region's GDP in the country's GDP;<br>Number of economic entities operating per 1000 inhabitants;<br>Number of bankrupt enterprises per 1000 operating companies;<br>Share of newly registered companies, compared to the operating entities;<br>Unemployment rate;<br>Share of exports of goods of the local origin in the region's GDP;<br>Revenue from export per capita;<br>Direct foreign investment per capita;<br>Material investment per capita;<br>Average gross monthly earnings;<br>Household savings per capita;<br>Ratio of the difference between the residents' income and the country's capital region;<br>Share of the working age population;<br>The population's domestic and international migration balance per 1000 inhabitants. |
| 2. GOVERNANCE EFFICIENCY CAPACITY<br>*2.1. Management efficiency*<br>*2.2. Financial possibilities* | Taxes paid and credited to municipal budgets per capita;<br>Municipal budget expenditure and income ratio;<br>General government gross debt compared to regional GDP;<br>Average annual ratio of recipients of social benefits to all residents;<br>Number of social risk families per 1000 inhabitants;<br>Pension beneficiaries per 1000 persons of working age. |
| 3. KNOWLEDGE AND INNOVATION CAPACITY<br>*3.1. Research and innovation development level*<br>*3.2. Innovation-encouraging environment* | Ratio of research and development (R&D) expenditure to GDP;<br>Expenditure on R&D in higher education and government sectors;<br>Share of corporate funds in the total R&D expenditure;<br>Added value created by the company;<br>Share of companies that implement innovations;<br>Export of high-tech goods, compared to overall exports;<br>Value added created in a company involved in professional, scientific, and technical activity, per employee;<br>Applications submitted to the European Patent Office (EPO) per 1,000,000 inhabitants;<br>Employees involved in R&D in the higher education and government sectors. |
| 4. LEARNING CAPACITY<br>*4.1. Learning system*<br>*4.2. Flexibility and competence of the labor market* | Number of people pursuing higher education (students at universities and colleges) per 1000 inhabitants;<br>Rate of lifelong learning (share of working population who attended training workshops in the recent year);<br>Share of population with higher education;<br>Average consumption expenditure per household member per month for education. |
| 5. SOCIO-ECONOMIC SYSTEM'S INFRASTRUCTURE<br>*5.1. The system of a modern and productive infrastructure*<br>*5.2. Sustainable development principles and pollution* | Households with internet access;<br>Percentage of people who used the internet every day in the last three months;<br>Companies using information technology (IT);<br>Road density (national and local);<br>Share of renewable energy in the total energy consumption;<br>Share of energy used in the GDP structure;<br>Emission of pollutants into the atmosphere from stationary sources of pollution in 1 km$^2$;<br>The amount of gases, causing the greenhouse effect, emitted into the atmosphere, by thousand t $CO_2$ equivalent;<br>Taxes paid and credited into municipal budgets for environmental pollution per capita. |

In order to ensure that the Resilio model indicators do not duplicate similar information (no redundant information) and trends, it is recommended to perform a collinearity analysis of selected indicators by calculating the Pearson linear correlation coefficients. If an indicator has a strong linear relationship with a major number of indicators used in the index, it is recommended to remove this indicator from the index. The indicators for which the Pearson linear correlation coefficient did not

exceed 0.8 remain in Table 1. Initially, 48 indicators were identified, but after the correlational analysis, 43 indicators remained (see Table 1).

The value normalization of indicators is performed in Phase 3. This is an obligatory action; however, indicators can be expressed by different units of measurement, and it is necessary to compare them with each other. In order to investigate an individual socio-economic system's development from a historical perspective and obtain it exactly, the following data normalization method was applied: the indicators for each Lithuanian region over the whole year were assessed and ranked only with the indicators of the pre-shock period (base year) of the same Lithuanian region. Such a method of data normalization allows the socio-economic system's corresponding years in the pre-shock year to be analyzed, i.e., to investigate the depth of vulnerability in comparison with the pre-shock year and/or when it recovered from the shock (recovery depth) to the pre-shock level. It is important to emphasize that, using such a data normalization method, each system is analyzed individually and the obtained information can be used from a historical perspective by analyzing and comparing how successfully the particular socio-economic system overcame economic shock over the relevant period.

After the data normalization, the calculation of Resindicis is performed (Phase 4). The indicators are placed into groups in the index based on the combined system indicators in Resilio.

$$\text{RESINDICIS} = (w_1) \text{ Ins\_Cap\_Resilio} + (w_2) \text{ Gov\_Cap \_Resilio} + (w_3)$$
$$\text{Inov\_Cap\_Resilio} + (w_4) \text{ Learn\_Cap \_Resilio} + (w_5) \text{ Inf\_Cap\_Resilio;} \tag{1}$$

$$\text{Ins\_Cap\_Resilio} = (w_5) \text{ Str\_Ins\_Resilio} + (w_6) \text{ Econ\_Vital \_Resilio;} \tag{2}$$

$$\text{Gov\_Cap \_Resilio} = (w_7) \text{ Gov\_Eff\_Resilio} + (w_8) \text{ Fin\_Opp\_Resilio;} \tag{3}$$

$$\text{Inov\_Cap\_Resilio} = (w_9) \text{ R\_Inov\_Resilio} + (w_{10}) \text{ Inov\_Env\_Resilio;} \tag{4}$$

$$\text{Learn\_Cap \_Resilio} = (w_{11}) \text{ Ed\_Syst\_Resilio} + (w_{12}) \text{ Lab\_Comp\_Resilio;} \tag{5}$$

$$\text{Inf\_Cap\_Resilio} = (w_{13}) \text{ Infrast\_System} + (w_{14}) \text{ Sustain\_Resilio.} \tag{6}$$

The definitions of the terms are listed below:

- RESINDICIS—the index of a socio-economic system's resilience to economic shocks;
- Ins_Cap_Resilio—insight capacity index;
- Gov_Cap_Resilio—socio-economic system's government capacity index;
- Inov_Cap_Resilio—knowledge and innovation index;
- Learn_Cap _Resilio—learning capacity index;
- Inf_Cap_Resilio—infrastructure capacity index;
- Str_Ins_Resilio—strategic insight sub-index;
- Econ_Vital _Resilio—economic vitality sub-index;
- Gov_Eff_Resilio—government efficiency sub-index;
- Fin_Opp_Resilio—financial opportunities sub-index;
- R_Inov_Resilio—research and innovation sub-index;
- Inov_Env_Resilio—innovation-friendly environment sub-index;
- Ed_Syst_Resilio—education system sub-index;
- Lab_Comp_Resilio—labor market flexibility and competence sub-index;
- Infrast_Sistem—a modern and productive infrastructional system sub-index;
- Sustain_Resilio—sustainability sub-index;
- wi—the coefficient of weight for determinant i.

In this stage, the coefficient of importance (weight) can be attributed to every determinant or group of determinants.

The additional stages may also include an index stability and sensitivity analysis, which substantiates index reliability and estimation transparency, and identification of the strengths and weaknesses of systems, the main strategic directions, and measures in order to increase the resilience.

Although estimations of the Resindicis index enable the comparison and ranking of systems by their resilience, the index itself does not reveal the degree of resilience. Considering the multifaceted nature of the concept of resilience and the fact that the degree of resilience can be increased by reducing vulnerability and/or by enhancing the ability to recover, the dynamic assessment of a system's resilience becomes extremely important. The degree of resilience will be revealed by the changes in pre-, over-, and post-shock resilience. The conceptual assessment of resilience is depicted in Figure 1.

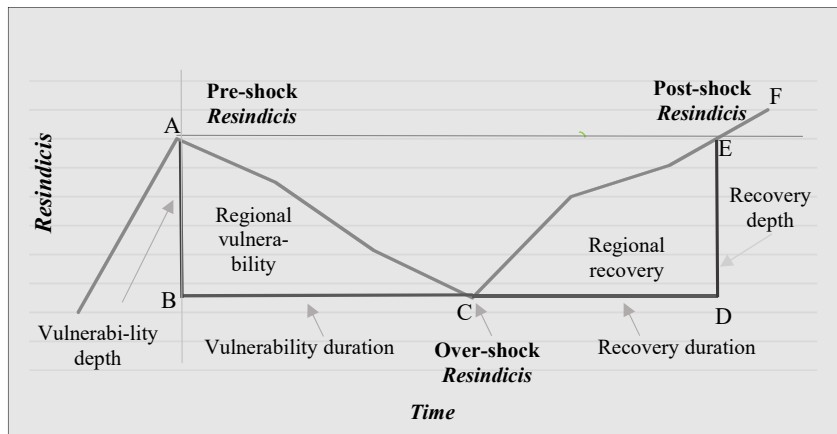

**Figure 1.** Conceptual principles of the assessment of resilience to economic shocks by Resindicis.

For the assessment of a system's resilience to an economic shock, the dynamics of the *Resindicis* index at the beginning and at the end of the shock (segment AB) are employed. The pre-shock Resindicis (point A) is a pivot point, from which the period of recovery (i.e., the period during which the post-shock Resindicis (point E) is achieved) starts being measured. Point C represents the over-shock Resindicis, while segment CD represents recovery duration, i.e., the period over which the pre-shock Resindicis matches up with the post-shock Resindicis. Segments AB and ED represent vulnerability and recovery depth, respectively. The vulnerability depth to recovery duration rate (line AB to line BC rate) and recovery depth to recovery duration rate (line ED to line CD rate) represent vulnerability and recovery speed, respectively. Areas ABC and CDE depict the vulnerability and recovery, respectively.

Because resilience to economic shocks is the ability of a system to remain as invulnerable as possible and/or to recover after an economic shock, it is estimated as the sum of area ABC (vulnerability) and area CDE (recovery). This means that a socio-economic system is more resilient to an economic shock when it is less vulnerable and more quickly reaches the pre-shock level. In this article, resilience is treated as a sum of vulnerability and recovery (see Figure 1). Vulnerability area ABC and recovery area CDE are estimated using the Resindicis curve change function in the corresponding segment.

$$R_i = p_i + a_i = \int_{tbs}^{tes} f_{ip}(t)dt + \int_{tes}^{tpp} f_{ia}(t)dt. \tag{7}$$

The definitions of the terms are listed below:

- R—resistance of the i-th system to an economic shock;
- t—period;
- tbs—the beginning of the shock;
- tes—the end of the shock;
- tpp—the post-shock period, when the pre-shock level is reached;

- $p\_i$—vulnerability area of the i-th system;
- $a\_i$—recovery area of the i-th system;
- $f\_ip$ (t)—vulnerability function of the i-th system;
- $f\_ia$ (t)—recovery function of the i-th system.

The beginning and the end of an economic shock are identified by considering the changes in selected socio-economic indicators (e.g., some scientists [42–44] recommend characterization of an economic crisis by the following indicators: a decrease in production volume (expressed as a decrease in GDP), a significant unemployment rate growth (expressed as an unemployment rate), a strong decline in living standards (expressed as a poverty risk rate), an increase in the state budget deficit (expressed as a government debt rate), and depreciation of a monetary unit's purchasing power (expressed as an inflation rate)). When the indicators start showing the trends of reduction (deterioration), this period is referred to as the beginning of an economic shock. The last period, for which the indicators start showing the trends of growth (improvement), is referred to as the end of an economic shock.

The above-introduced function of resilience shows that a socio-economic system is more resilient to economic shocks when it is less vulnerable and more quickly reaches the pre-shock level. This means that the i-th system is more resilient when the sum of its vulnerability area $p_i$ and recovery area $a_i$ is lower in comparison to the sums of vulnerability and recovery areas estimated for other systems. It should be noted that the identification of the most resilient system is based on a consideration of vulnerability and recovery area sums, i.e., the most resilient system's vulnerability and recovery areas do not have to be smallest as separate objects in the context of other systems, but the sum of both areas has to be the lowest. Hence, the identification of the system which is most resilient to economic shocks in comparison to other systems is based on the following function (developed by the authors):

$$R_{mostresilient} = {}_{min}(R_i), \text{ when i varies from } 1 \text{ to } n. \tag{8}$$

The above-introduced function, which enables the identification of the most resilient system, proposes that most resilient systems are not necessarily hardly vulnerable and quickly recovering ones; some of them are indeed characterized by high vulnerability levels, but at the same time they are able to recover relatively quickly, while others are hardly vulnerable, but able to recover only in the relatively long term. Because resilience is treated as a continuous process, the situation in which vulnerability and recovery areas, estimated for the most resilient system, are not smallest helps identify the strategic weaknesses of this system (the conceptual principles of the Resindicis assessment are applied). In other words, if the vulnerability of the most resilient system is relatively high (when the recovery rate is high), vulnerability reduction should be treated as a priority, but if recovery is relatively slow (when the vulnerability rate is low), priority should be given to recovery promotion.

## 4. Results

To verify the practical applicability of the methodology developed for the assessment of a socio-economic system's resilience to economic shocks introduced in this article, an empirical research of 10 Lithuanian counties (regions at the NUTS III level (in the empirical analysis, the socio-economic system is equated to the county (region) of Lithuania)) and the example of the 2008 global financial crisis was conducted. The analysis period was 2006–2016.

A small and open Lithuanian economy cannot avoid the impact of shocks in global finance markets, as proven by some historical facts; the greatest economic declines in Lithuania after restoration of the country's independence were caused by some external factors—the 1998 financial crisis in Russia and the 2008 global financial crisis, which began in September of that year after the collapse of "Lehman Brothers". The February 2018 events in the United States' stock market, along with the threat of real-estate bubbles in the world's major finance centers and the growing imbalance of the global economy, raise concerns about a potentially new financial shock. This is why it is very important to

review the lessons of previous crises and assess the resistance of Lithuanian regions to a potentially new economic shock.

Lithuanian regions are dissimilar in both social and economic development. With reference to the White Paper of Lithuanian Regional Policy 2017–2030 [61], Lithuanian regions are attributed to three hierarchical levels. The three largest Lithuanian counties (Vilnius, Kaunas, and Klaipeda) are ranked as the regions of European importance. They bring together top-notch services and special infrastructures (universities, research and development (R&D) infrastructure, university hospitals, etc.), and provide support for the development of high-value-added services. Two medium-sized counties (Siauliai and Panevezys) are ranked as the regions of national importance. They bring together university departments and region-specific state-owned enterprises and institutions, and provide the support for the development of high-value-added services and production. The remaining five counties (Alytus, Marijampole, Taurage, Telsiai, and Utena) are ranked as the centers of regional importance. They bring together vocational training, colleges and their departments, territorial departments of business service institutions, region-specific departments of state-owned enterprises, and the secondary level of healthcare, and they provide the support for the development of high- and medium-value-added production.

It should be noted that the research of practical applicability of the methodology covers all 10 Lithuanian regions; however, due to the limited scope of this article, the result graph depicts only three of them—the Vilnius, Kaunas, and Klaipeda regions. The decision to visualize the results estimated for the regions of European importance was determined by the fact that the population in the Vilnius, Kaunas, and Klaipeda regions amounts to nearly 58 percent of the total population in Lithuania; the abovementioned regions create nearly three-quarters (i.e., 75 percent) of the country's GDP, and the export from these regions amounts to nearly 50 percent of the total exports of Lithuanian origin. Thus, the economic vulnerability and resilience of these regions has a significant impact on Lithuanian economics at both the national and international level.

Based on the historical socio-economic indicators of Lithuanian regions, it can be stated that the economic decline in Lithuania began in 2008 (foreign direct investment (FDI) rates reacted the most sensitively; later on, the fluctuations in GDP rates could be observed). Nevertheless, it should not be overlooked that different indicators showed their peaks and improvement trends at different periods of time. For instance, GDP and FDI started growing in 2010, when unemployment rate reached its highest value. These findings confirm that the assessment of these regions' resilience by employing a single indicator is inexpedient due to the lagged effects. On the contrary, the assessment of any socio-economic system's resilience should be based on a complex approach.

The vulnerability and recovery indices estimated for the European-level Lithuanian regions for the period of 2006–2014 are presented in Figures 2–4.

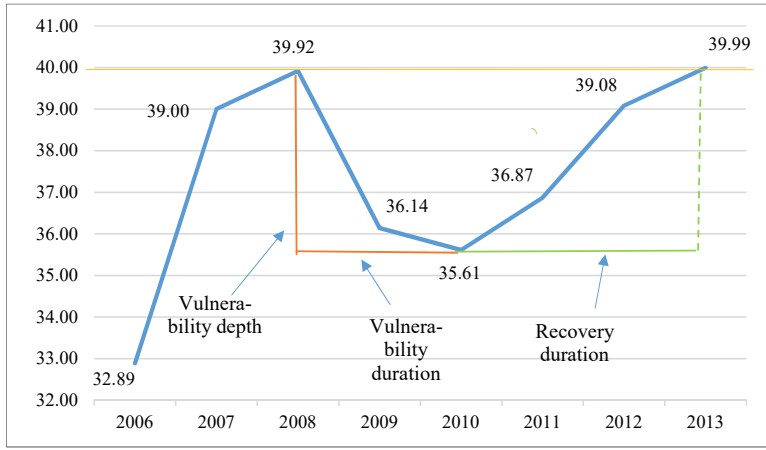

**Figure 2.** Vulnerability and recovery indices estimated for the Vilnius region for the period of 2006–2013.

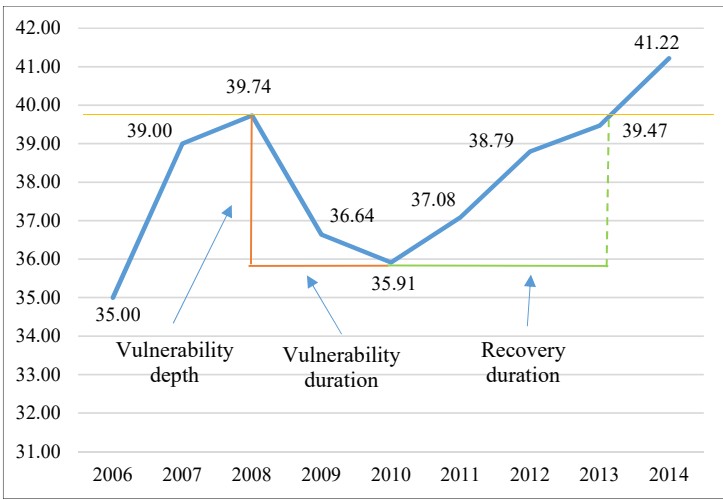

**Figure 3.** Vulnerability and recovery indices estimated for the Kaunas region for the period of 2006–2014.

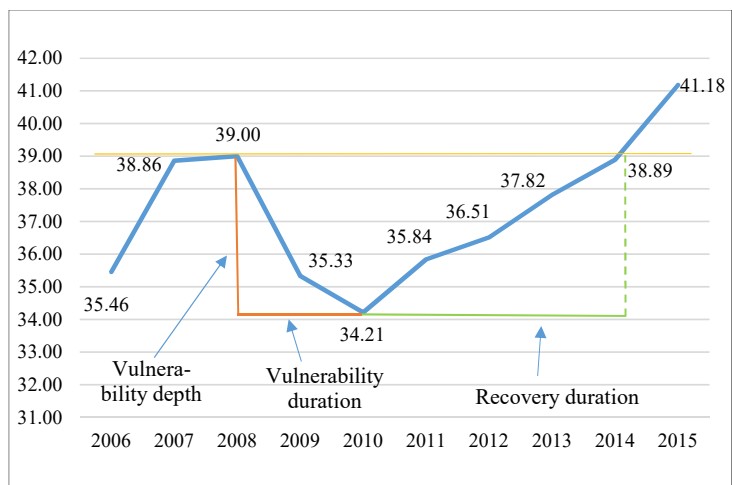

**Figure 4.** Vulnerability and recovery indices estimated for the Klaipeda region for the period of 2006–2014.

The summary of the vulnerability and recovery indicators estimated for all Lithuanian regions is presented in Table 2.

Vulnerability indicates that a socio-economic system is poorly protected from economic shocks. The fact that the global financial crisis began in 2008, and, at the same time, Lithuanian regions became vulnerable (Lithuanian economics reacted to the economic shock) confirms the finding that the analyzed socio-economic systems are relatively sensitive to economic shocks.

The research also reveals that different Lithuanian regions were vulnerable for an unequal period of time. The vulnerability of the largest part of the regions lasted for two years (from 2008 to 2010), except for the vulnerability of the Alytus and Utena regions, which lasted until 2011. Furthermore, different regions suffered from unequally serious damage caused by the economic shock. The maximum vulnerability depth and speed (vulnerability speed is estimated as the ratio of vulnerability depth to vulnerability duration; recovery speed is estimated as the ratio of recovery depth (equal to vulnerability depth) to vulnerability duration) were captured for the Siauliai and Panevezys regions, while the minimum values were for the Utena, Kaunas, and Taurage regions.

**Table 2.** Vulnerability and recovery depth, duration, and speed estimated for Lithuanian regions for the period of 2007–2016.

| Criterion | Regions | | | | | | | | | | Average |
|---|---|---|---|---|---|---|---|---|---|---|---|
| | European (I) Level | | | National (II) Level | | | Regional (III) Level | | | | |
| | Vilnius | Kaunas | Klaipeda | Siauliai | Panevezys | Telsiai | Alytus | Utena | Marijampole | Taurage | |
| Vulnerability depth (indices) | 4.31 | 3.82 | 4.79 | 6.25 | 5.07 | 4.26 | 4.96 | 3.08 | 4.53 | 3.96 | 4.50 |
| Vulnerability duration (years) | 2.00 | 2.00 | 2.00 | 2.00 | 2.00 | 2.00 | 3.00 | 3.00 | 2.00 | 2.00 | 2.20 |
| Vulnerability speed (coefficient) | 2.15 | 1.91 | 2.39 | 3.13 | 2.54 | 2.13 | 1.65 | 1.03 | 2.26 | 1.98 | 2.12 |
| Vulnerability | 2.69 | 2.50 | 3.52 | 4.86 | 5.52 | 4.66 | 3.50 | 4.94 | 5.54 | 5.01 | 4.17 |
| Recovery duration (years) | 2.92 | 3.15 | 3.57 | 5.63 | 3.59 | 1.95 | 4.48 | 4.78 | 3.46 | 5.02 | 3.86 |
| Recovery speed (coefficient) | 1.47 | 1.21 | 1.34 | 1.11 | 1.41 | 2.18 | 1.11 | 0.64 | 1.31 | 0.79 | 1.26 |
| Recovery | 6.59 | 3.15 | 7.26 | 20.72 | 11.63 | 5.44 | 10.70 | 8.02 | 11.05 | 11.17 | 9.37 |
| Resilience (vulnerability + recovery) | 9.27 | 5.65 | 10.77 | 25.58 | 17.15 | 10.10 | 14.20 | 12.96 | 16.58 | 16.18 | 13.54 |
| Recovery/vulnerability duration ratio | 1.46 | 1.58 | 1.79 | 2.81 | 1.79 | 0.97 | 1.49 | 1.59 | 1.73 | 2.51 | 1.77 |

The results of this research indicate that Lithuanian regions were insufficiently prepared for the economic shock of 2008. On average, it took nearly 1.7 times longer for Lithuanian regions to recover after the 2008 shock than to suffer from the shock exposure. Only for the Telsiai region did the recovery duration take longer than the duration of shock exposure. The vulnerability speed of all regions was faster than the recovery speed. In summary, it can be stated that the duration of the recovery after an economic shock for the analyzed socio-economic systems took longer than the duration of vulnerability. Thus, the conclusions of the empirical research highlight the necessity to develop and implement a socio-economic system's resilience strategies and measure plans.

Resilience is a socio-economic system's ability to resist, absorb, and withstand an economic shock. For this reason, vulnerability and recovery indicators must be separately estimated for every system. Telsiai, Kaunas, and Vilnius regions were most resilient to the economic shock of 2008 in Lithuania. However, at the same time, these regions were strongly vulnerable, except for the Telsiai region, which is characterized by one dominating factor (the export of petroleum products), determining the reduced vulnerability and fast recovery of the region. This tendency suggests the hypothesis that specialization may determine its resilience, but this hypothesis can only be confirmed or rejected after more comprehensive research. Despite the fact that the Vilnius and Kaunas regions were severely affected by the economic shock of 2008, the recovery of these regions was fast, which, in turn, determined a relatively high degree of resilience.

## 5. Analysis and Discussion

The empirical research in the context of Lithuanian regions revealed the following tendencies:

- Socio-economic systems of different development levels are unequally affected by an economic shock;
- Lithuanian regions' recovery from the global economic shock in 2008 lasted longer than their vulnerability;
- Lithuanian regions' speed of vulnerability was faster than the speed of their recovery;
- The resilience of the economically developed Lithuanian regions was determined by their ability to recover, rather than by their ability to stay less vulnerable; such systems were characterized not only by great vulnerability depth, but also by fast recovery;
- Less economically developed Lithuanian regions showed longer recovery terms.

The empirical estimations disclosed that both vulnerability and slow recovery are the weaknesses of Lithuanian regions. A high degree of vulnerability is a serious problem of the strongest Lithuanian regions, while the other regions are not only vulnerable, but also show extremely low recovery rates. Although the regions which occupy lower positions by Resindicis showed lower vulnerability rates than the regions with higher Resindicis, weak economies in the former determined that they suffered greater losses even from relatively slight economic downturns in comparison to the regions with stronger economies. Considering that Lithuania is a small economy, where domestic consumption is insufficient for economic development, and where export is vital for the development of regional economies, the efforts to create efficient resilience promotion strategies should be directed toward vulnerability reduction and the search for opportunities to accelerate recovery after economic shocks. The data in Table 2 (i.e., the instant information on the resilience of particular Lithuanian regions to the economic shock of 2008) confirm that Lithuanian regions were not ready for the economic shock of 2008. These data can be considered as the primary information for the assessment of the ability of regions to avoid, resist, and withstand any economic shock.

This article presents variations in certain aspects of the evaluation methodology that provide important information needed to formulate promotion strategies and implementation measures to enhance the resilience of an individual socio-economic system. Resindicis and the analysis of vulnerability enabled the classification of Lithuanian regions into four groups (see Figure 5) and the identification of similar regions (competitors).

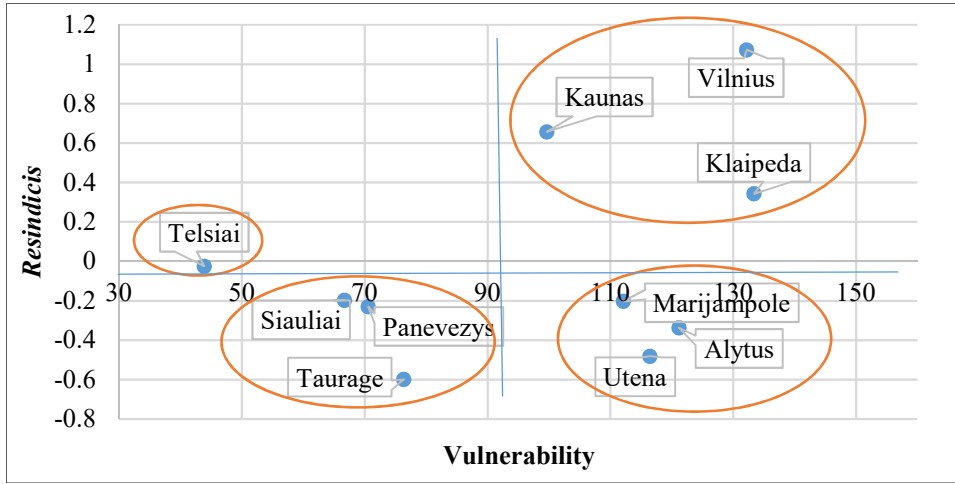

**Figure 5.** Relationship between the Resindicis of Lithuanian regions and their vulnerability.

Group I is characterized by strong resilience and low vulnerability. The group features only the Telsiai region (a socio-economic system with a regional specialization at the national level).

Group II is characterized by strong resilience and high vulnerability. The group combines three socio-economic systems: the Vilnius, Kaunas and Klaipeda regions (European-level regions).

Group III is characterized by weak resilience and low vulnerability. The group combines the Siauliai and Panevezys regions (national level regions) and the Taurage region (regional level region).

Group IV is characterized by weak resilience and high vulnerability. The group combines three socio-economic systems: the Marijampole, Utena, and Alytus regions (regional level regions).

The empirical study showed that the economies of the Lithuanian regions were affected and took relatively long to recover after the 2008 global economic crisis and its consequences. Although Lithuania's regions are yet to prepare individual strategies for each region in order to increase its resilience and individual measures are yet to be included in the regional strategic development plans, the Lithuanian government is already taking some measures in order to ensure and increase the resilience of the country both at the national level and that of each Lithuanian region individually. Since 2018, the Lithuanian government began implementing six structural reforms in the fields of education, health, tax, innovation, anti-shadow economy, and pensions. All these reforms are aimed at ensuring the geographically balanced and sustainable economic growth and quality of life throughout Lithuania. Although no modeling of the impact of these structural reforms on the resilience of Lithuanian regions was done so far (this could be a field of further research), their positive impact is more than expected; education reform will directly affect learning capacities; health reform will affect economic viability; and tax, anti-shadow economy, and pension reforms will affect knowledge and innovation capacity.

In order to evaluate the reliability of the assessment methodology for resilience to economic shock, a robustness and sensitivity analysis was performed, in which the results of the calculations were compared using (a) different data normalization methods, (b) different factor selection methods, and (c) different weight coefficients for resilience-determining capacities.

The impact of the possible change in the Resindicis calculation methodology on the overall result was verified by raising the hypothesis of the Kendall concordance coefficient equal to zero. The chosen significance level for the hypothesis was $\alpha = 0.05$. The hypothesis of the zero-equality coefficient was rejected when the observed *p*-value was estimated to be less than 0.05.

The compatibility of the obtained results indicates that the accuracy of the assessment most strongly depends on the following methods (presented in decreasing order): (1) giving the weight coefficients for the sub-indices, (2) the selection of factors and indicators determining resilience, and (3) normalization of data. Tables 3 and 4 illustrate the impact of weight coefficient selection on the final result.

**Table 3.** The Resindicis of the Lithuanian regions, using different weight coefficient methods, ranking the assessment compatibility for different resilience of Lithuanian regions.

| Compatibility of Different Weight Coefficients Methods for Different Lithuanian Regions, 2007–2015 | W | *p*-Value |
|---|---|---|
| Regions: most and least resilient (Vilnius, Kaunas, Klaipeda, Telsiai, Taurage) | 0.9898 | 0 (<0.05) |
| Regions: satisfactorily resilient (Marijampole, Panevezys, Siauliai, Utena, Alytus) | 0.7516 | 0 (<0.05) |

**Table 4.** The Resindicis of the Lithuanian regions, using different weight coefficient methods, ranking the assessment compatibility in different periods of time.

| Compatibility of Different Weight Coefficients Methods | W | *p*–Value |
|---|---|---|
| Period: 2007–2015 years | 0.9028 | 0 (<0.05) |
| Pre-shock and post-shock period: 2007, 2011–2017 years | 0.9389 | 0 (<0.05) |
| Over-shock period: 2008–2010 years | 0.8681 | 0 (<0.05) |

The data in Table 3 highlight the tendency that weight coefficients methods mostly work according to Resindicis satisfactorily resilient regions, because the values of Kendall's concordance coefficient were less close to 1.

The compatibility of different weight coefficient methods, ranking assessment in different periods of time, shows that the ranks are more sensitive to giving weight coefficients in the over-shock period in comparison with the pre-shock or post-shock period, because, in the over-shock period, each region responds to economic shock in a different way (vulnerability depth and duration are different).

A robustness and sensitivity analysis confirmed that, for assessing a socio-economic system's resilience to economic shock, it is recommended to use the standard deviation from the mean and distance from the minimum and maximum values as data normalization methods. Resindicis is especially sensitive to the factors and indicators determining methodology for the Lithuanian regions, which are exceptionally distinguished from the national average according to the relevant indicators.

It is important to note that, if the methodology developed for the assessment of a socio-economic system's resilience to economic shocks introduced in this article were to be based on a different data normalization method—i.e., if the indicators which characterize the resilience of every system included in the Resilio index were compared with the corresponding national average in the base year—this would provide the opportunity to rate socio-economic systems by their vulnerability and recovery in the context of the entire group of socio-economic systems (the regions, or the entire country). Such a data normalization method would also enable researchers to establish how deeply a system was damaged in comparison to the national average and when (or whether) it was able to restore itself to its pre-shock condition.

If a socio-economic system does not reached its pre-shock condition over the period under research, the methodology introduced in this article enables a prediction of the recovery (by making the regression equation from the Resilio estimated for the periods available). Nevertheless, it must be noted that the exact estimation of when a system could recover and reach its pre-shock condition is complicated because it largely depends on how well this system is prepared for recovery and what strategic actions it takes. The impact of the external environment (international and national policies, economics, global trends, potential economic shocks, etc.) is no less important and must be considered when predicting a possible period of recovery.

The empirical research justified that the methodology developed for the assessment of a socio-economic system's resilience to economic shocks is an appropriate tool for the assessment of resilience, economic analysis, and the effectiveness of resilience promotion strategies.

## 6. Conclusions

In the research into socio-economic systems' resilience, it is important to identify the impact of an economic shock on the system's objects and to investigate how the development of this system is affected by relevant economic shocks. The impact of economic shocks on a socio-economic system's development can be bidirectional: either negative (may damage the economies) or positive (economic shocks may provide new opportunities for the socio-economic system's development). Economic shocks may directly or indirectly affect both the subjects of the socio-economic system and the entire economy during pre-, over-, and after-shock (i.e., until the time when the pre-shock condition is reached) periods.

Consideration of a single factor does not disclose the problems of a socio-economic system's resilience to economic shocks. The fact that an economic shock affects all subjects of the system, whereby a socio-economic system is treated as a structured set of entities and objects interacting in a specific geographic area (country, region, or town), justifies a complex assessment of the resilience concept. The following methodological peculiarities of the assessment of a socio-economic system's resilience to economic shocks were identified:

- One or several indicators of resilience inadequately reflect the characteristics of resilience to economic shocks; therefore, its resilience to economic shocks should be assessed by employing a complex set of indicators;
- The methodological substantiation of resilience indicators and capacities contributes to the reliability of the assessment;
- Identification of the specific indicators and capacities of resilience ensures a higher accuracy of assessment;
- Incorporation of the indicators of resilience in the index estimation methodology substantiates the assessment of its resilience by an index;
- The analysis of a socio-economic system's vulnerability and recovery, included in the assessment of its resilience, illustrates its ability to stay resilient from a time perspective.

The research on the resilience of Lithuanian regions to economic shocks revealed that the weak spot of the Lithuanian regions lies in both their vulnerability and recovery. A high degree of vulnerability is a serious problem of the main (economically strongest at the European level) Lithuanian regions, while the other regions (at the national and international levels) are not only vulnerable, but also show extremely low recovery rates.

The hypothesis that the specialization of the socio-economic system determines its resilience requires additional research.

The indicator correlation analysis, included in the index calculation, enables the elimination of the interrelated indices. Thus, the duplication of similar information is eliminated, as well as making Resindicis more convenient for practical use.

The empirical research disclosed the advantages and disadvantages of the methodology developed for the assessment of a socio-economic system's resilience to economic shocks. The main advantages of this methodology are as follows: (a) it allows for the complex assessment of the resilience of a particular system to economic shocks in comparison to other systems and in terms of time; (b) it incorporates not only quantitative, but also qualitative indicators (by estimation of weight coefficients); (c) it conveniently (in a single number) expresses the rate of resilience; (d) it enables the identification of the most resilient and most vulnerable socio-economic systems and the system with the highest recovery speed; (e) it provides the opportunity to rate socio-economic systems by their vulnerability and recovery in both the context of the entire group of systems (or the entire country) and the context of the development of a single socio-economic system; (f) it enables a prediction of the period of a system's recovery; (g) it enables the assessment of resilience from static (i.e., with consideration of resilience determinants and capacities) and dynamic (i.e., with consideration of the changes in the Resindicis index in comparison to the value of the index in the base year) perspectives. The main

drawbacks of the methodology are as follows: (a) it does not consider the risk of the emergence of potentially new economic shocks; (b) if economic shocks are short-term, the prognostication of the period of recovery is relatively inaccurate due to the small number of periods employed. The practical application of the presented methodology, based on the examples of other countries and regions, could be a future research area.

**Author Contributions:** J.B. conceived the idea and together with I.P. wrote and revised the manuscript; O.P., Z.S. organized and performed the data analyses and interpreted data. All authors approved the final manuscript.

**Funding:** This research received no external funding.

**Acknowledgments:** The authors would like to thank the editors and anonymous reviewers for their constructive comments and helpful suggestions

**Conflicts of Interest:** The authors declare no conflict of interest.

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
