# Peer review of "An Assessment of Socio-Economic Systems’ Resilience to Economic Shocks: The Case of Lithuanian Regions"

_sustainability, doi:10.3390/su11030566_

Reviewer 1 Report

The addition of the term 'socio-economic' throughout the document, sometimes several times in one sentence, is not conducive to more clarity. It is simply overkill. This should be corrected through language editing.

I keep wondering why the authors try to make a point about 'regional systems' when the unit of analysis is Lithauania, a nation state. I do not see the point. There is nothing wrong with identifying a nation state as a 'system'. In my opinion the paper, particularly the Introduction and parts of the conceptual framework (section 2), can be shortened by deleting the unneccessary elaborations on 'regional' system. In this vein, footnote 1 on p. 11 is redundant.

Author Response

The term 'socio-economic' throughout the article, which are repeating in one sentence or can be eliminated, without losing the idea of the sentence, are deleted.The term of regional or regions, in the theoretical part are changed to the term system.

The name of the article was specified, according to the remarks: An assessment of socio-economic systems’ resilience to economic shocks: the case of Lithuanian regions

The relationship between regional system and national system are corrected in the article. In the theoretical and methodological part of the article the socio-economic system approach is left. In the empirical part – the socio-economic system is treated as Lithuanian regions.

The theoretical model of factors (Resilio) is universal and can be applied to the evaluation of the resilience of the socio-economic system of different levels (groups of countries (for example European Union, the Baltic sea Region), country, region (county), city). Empirically, the model was adapted to the region of the country (10 Lithuanian counties). In the article, Lithuania is not analyzed as a national state, but as a socio-economic system, that consists of 10 socio-economic sub-systems (counties), it est regions inside the country Lithuania.

The footnote 1 on p. 11 is deleted.

The structure of the article was corrected, according to the suggestions.

Reviewer 2 Report

The paper is quite broad and in-depth. However it is very long in the part related to the "Materials and Methods" that I would rather divide in the "Backgroung of the literature" and "Methodology" used. The analysis of the situation in the Lithuanian regions seems to me more used to test the validity of the proposed model than to analyze the socio-echnomic context of Lithuania. However, the research seems really interesting and the subject of subsequent investigations.

Author Response

The structure of the article was corrected, according to the suggestions:

1. Introduction

2. Literature Review

2.1. The concept of socio-economic systems’ resilience to economic shocks

2.2. The main factors of socio-economic systems’ resilience to economic shocks

2.3. Socio-economic system’s resilience to economic shocks assessment methods

3. Data and Methodology

4. Results

5. Analysis and Discussion

6. Conclusions

The term 'socio-economic' throughout the article, which are repeating in one sentence or can be eliminated, without losing the idea of the sentence, are deleted.

The name of the article was specified, according to the remarks: An assessment of socio-economic systems’ resilience to economic shocks: the case of Lithuanian regions

The relationship between regional system and national system are corrected in the article. In the theoretical and methodological part of the article the socio-economic system approach is left. In the empirical part – the socio-economic system is treated as Lithuanian regions.

Reviewer 3 Report

I reccommend justifing because the econometric methods is better than other possible methods.

Author Response

The focus in the article was put on the factors determining the resilience of the socio-economic system, to the evaluation method by the index, and to the individual components of the resilience - vulnerability and recovery. The robustness and sensitivity analysis (from L 705) confirmed the reliability of the method used in the article. The purpose of this article was to identify the main factors of socio-economic systems’ resilience to economic shocks and to use the index evaluation method for the assessment of socio-economic systems’ vulnerability and recovery from the economic shock. The results obtained in the article with the help of the econometric methods can be used to analyze the concept of the resilience in further researches.

Reviewer 4 Report

The paper ‘An assessment of socio-economic systems’ resilience to economic shocks: the case of Lithuania’ is without any doubt an interesting paper that tries to clarify several concepts related to resilience and vulnerability.  Nonetheless, the paper fails quiet shortly on this regard. In fact the main shortcoming of the work is its organization and structure.

First I would like to see a dedicated section to the concepts of factors behind socio-economic systems’ resilience (e.g. I will extend the section 2.1 in a new section 2). However, many articles related to this work (that actually are on line with the present paper) are not cited. Please refer to [1] and [2] for the reviews of the literature on resilience and vulnerability that are in lines with this paper.  Please refer to [3] for a framework that link vulnerability and resilience, especially on the light of natural disaster assessment (that you mention in the paper) or alternatively you can cut that part. Please refer to [4] for alternative construction of resilience indicators.  Please refer to [5] and [6] for study of economic resilience at European level .

Second, the role of vulnerability is quite relevant in you paper. It deserves more attention and you can cite this concept since the title.

Finally, there is the need to provide some policy implications for the Lithuanian case

References

[1]  Modica, M., & Reggiani, A. (2015). Spatial economic resilience: overview and perspectives. Networks and Spatial Economics15(2), 211-233.

[2] Modica, M., Reggiani, A., & Nijkamp, P. (2018). Vulnerability, resilience and exposure: methodological aspects and an empirical applications to shocks (No. 1318). SEEDS, Sustainability Environmental Economics and Dynamics Studies.

[3] Modica, M., & Zoboli, R. (2016). Vulnerability, resilience, hazard, risk, damage, and loss: a socio-ecological framework for natural disaster analysis. Web Ecology16(1), 59-62.

[4] Graziano, P., & Rizzi, P. (2016). Vulnerability and resilience in the local systems: The case of Italian provinces. Science of the Total Environment553, 211-222

[5] Modica, M., & Reggiani, A. (2014, November). An alternative interpretation of regional resilience: evidence from Italy. In ERSA conference papers (No. ersa14p369). European Regional Science Association.

[6] Cainelli, G., Ganau, R., & Modica, M. (2018). Industrial relatedness and regional resilience in the European Union. Papers in Regional Science.

Author Response

The structure of the article was corrected, according to the suggestions:

1. Introduction

2. Literature Review

2.1. The concept of socio-economic systems’ resilience to economic shocks

2.2. The main factors of socio-economic systems’ resilience to economic shocks

2.3. Socio-economic system’s resilience to economic shocks assessment methods

3. Data and Methodology

4. Results

5. Analysis and Discussion

6. Conclusions

 The suggested articles were included in the article.

The concepts and relationship between resilience and vulnerability are discussed additionally (from line 108 and 160)

The review of policy implication in Lithuania is added in the article (from line 700).

The terms 'socio-economic' throughout the article, which are repeating in one sentence or can be eliminated, without losing the idea of the sentence, are deleted.

The name of the article was specified, according to the remarks: An assessment of socio-economic systems’ resilience to economic shocks: the case of Lithuanian regions

The relationship between regional system and national system are corrected in the article. In the theoretical and methodological part of the article the socio-economic system approach is left. In the empirical part – the socio-economic system is treated as Lithuanian regions.

Round  2

Reviewer 4 Report

I am satisfied with the changes  provided by the authors